# Influence of Obesity and Fluid Balance on Operative Outcomes in Hepatic Resection

**DOI:** 10.3390/jpm12111897

**Published:** 2022-11-13

**Authors:** Suk-Won Suh

**Affiliations:** Department of Surgery, Chung-Ang University College of Medicine, Chung-Ang University Hospital, 224-1, Heuk Seok-Dong, Dongjak-Ku, Seoul 156-755, Korea; bumboy1@cau.ac.kr; Tel.: +82-2-6299-3184; Fax: +82-2-824-7869

**Keywords:** obesity, blood loss, central venous pressure, fluid restriction, hepatic resection

## Abstract

As the number of obese patients requiring hepatic resection is increasing, efforts to understand their operative risk and determine proper perioperative management are necessary. A total of 175 patients who underwent hepatic resection between March 2015 and July 2021 were evaluated. The patients were divided into two groups by their body mass index (BMI) using the World Health Organization definition of obesity for Asians: obese patients (BMI ≥ 25 kg/m^2^, n = 84) and non-obese patients (BMI < 25 kg/m^2^, n = 91). The operative duration (195.7 ± 62.9 min vs. 176.0 ± 53.6 min, *p* = 0.027) was longer and related to a higher estimated blood loss (EBL) ≥ 500 mL (61.9% vs. 40.7%, *p* = 0.005) in the obese patients than in the non-obese patients. Obesity (odds ratio (OR), 2.204; 95% confidence interval (CI), 1.177–4.129; *p* = 0.014) and central venous pressure (CVP) ≥ 5 (OR, 2.733; 95% CI, 1.445–5.170; *p* = 0.002) at the start of the surgery were significant risk factors for EBL ≥ 500 mL. Obese patients with low CVP showed significantly lower EBL than those with high CVP, but a similar EBL to non-obese patients (*p* = 0.003). In conclusion, fluid restriction before hepatic resection would be important, especially in obese patients, to improve their operative outcomes.

## 1. Introduction

Recent advances in perioperative care, surgical techniques, and dissection devices expand the indications of hepatic resection including advanced HCC and metastatic tumor in the liver [1,2]. The wide spread of hepatocellular carcinoma (HCC) surveillance in high-risk populations, such as cirrhotic patients, also has led to an increase in the detection of early-stage HCC that is eligible for hepatic resection [3]. Along with these increments, the parallel increase in the prevalence of obesity was reported in populations across the world in recent decades. The WHO defines obesity as a BMI above 30 kg/m^2^; however, the cut-off point for obesity is low for Asians considering the prevalence and incidence of metabolic disease, including cardiovascular diseases, dyslipidemia, and DM, which varies greatly among ethnic groups. An obesity epidemic has been observed in many industrialized countries, including South Korea, with the recent literature reporting a prevalence of 35.7% in 2018 [4]. Some obese people have nonalcoholic steatohepatitis (NASH), which can progress to liver cirrhosis and, subsequently, to the development of HCC [5]. These findings indicate the likelihood of an increase in obese patients requiring hepatic resection. 

Obesity has been shown to increase the risk of excessive blood loss during hepatic resection in previous studies [6,7]. This might be life-threatening and result in poor early postoperative outcomes, as well as tumor recurrence in the long-term, related to subsequent transfusion requirements [8,9]. It is currently thought that patients with a high BMI require higher ventilation pressure, which could impair hepatic venous outflow [10,11]. Hepatic blood congestion leads to an incremental increase in transmural pressure and the distension of hepatic veins, which are consequently easily torn, causing blood loss at the time of parenchymal transection [12]. This bleeding might be troublesome, and is not stopped by portal triad clamping because of the different vascular outflow system of the liver. In addition, obese patients have hepatic steatosis with or without steatohepatitis, which causes the liver tissue to be more frangible and promotes blood loss during hepatic resection [13]. Therefore, maintaining a low pressure in the central venous system, which is correlated with low pressure in the hepatic vein and liver sinusoids, would be more important in obese patients to minimize intraoperative blood loss during hepatic resection. 

The aim of the present study was, first, to investigate the influence of obesity on operative outcomes of patients after hepatic resection and, second, in a group of obese patients, to determine whether preoperative fluid restriction for reducing hepatic venous pressure can improve clinical outcomes by minimizing intraoperative blood loss.

## 2. Patients and Methods

### 2.1. Patients

A total of 217 patients who underwent hepatic resection between March 2015 and February 2022 at our hospital were evaluated. Patients with upper abdominal surgical history (n = 8), combined operation (n = 13), treatment history of hepatic lesion (n = 7), borderline liver function (n = 6), preoperative renal dysfunction (n = 5), and insufficient clinical data (n = 3) were excluded from the analysis. Finally, 175 patients were enrolled in the study. Patients were stratified into two groups based on the BMI according to the Korean Society for the Study of Obesity and a World Health Organization (WHO) definitions of obesity for Asian [4,14]: an obese group (BMI ≥ 25 kg/m^2^, n = 84) and a non-obese group (BMI < 25 kg/m^2^, n = 91). Demographics and operative outcomes were compared between the groups. Further, in an obese group, patients were divided into two groups by their central venous pressure (CVP) at start of surgery and differences in clinical outcomes compared to those of a non-obese group were investigated: obesity with a CVP < 5 mmHg (group A, n =31); obesity with a CVP ≥ 5 mmHg (group B, n = 53); non-obese patients (group C, n = 91).

### 2.2. Data Collection

Clinico-demographic data, including age, sex, body mass index, presence of diabetes mellitus (DM), hypertension (HTN), diagnosis and preoperative laboratory results of liver function, including total bilirubin (TB), international normalized ratio (INR), and albumin, were collected. The type of hepatic resection was classified into major resection, defined as the resection of three or more segments, such as right hemi-hepatectomy, extended right hemi-hepatectomy, left hemi-hepatectomy, extended left hemi-hepatectomy, and central hepatectomy, or minor resection. Operative data, including operative duration, changes in CVP during hepatic resection, amounts of fluid administered and urine output, vasopressor usage, requirement of blood transfusion, estimated blood loss (EBL), were investigated. Data regarding the nature and incidence of postoperative complications and intensive care unit (ICU) admission, mortality and length of postoperative hospital stay were also collected. Perioperative laboratory results of liver function (TB, INR, and albumin) and renal function (creatinine) were analyzed. In addition, all available intake records, composed of oral and parenteral fluids and output data, including urine, gastrointestinal losses, and drains, from operative day to postoperative day 7, were collected. Acute kidney injury (AKI) was defined in accordance with the 2012 Kidney Disease Improving Global Outcomes guidelines [15], which had higher predictability than other criteria for assessing prognosis: increase in serum creatinine by ≥ 0.3 mg/dL within 48 h; increase in serum creatinine to ≥1.5 times the baseline within 7 days before surgery; or urine volume < 0.5 mL/kg/h for 6 h. Postoperative liver insufficiency was defined as a peak postoperative TB level of >7 mg/dL and/or the presence of ascites > 500 mL/day based on a previous study [16]. 

### 2.3. Anesthetic and Surgical Technique

All patients underwent ultrasonography-guided right internal jugular vein catheterization after tracheal intubation in the operating room, and the position of the catheter was determined using a chest radiograph. Electrocardiogram, pulse oximetry, end-tidal carbon dioxide, invasive radial arterial pressure, CVP, and urine output were monitored. Fluid was not administered preoperatively and was restrictively infused after the start of anesthesia, maintaining a CVP of less than 5 mmHg until the hepatic parenchymal transection was complete. Thereafter, 1000–1500 mL/h of crystalloid and colloid solutions were compensatory administered in the operating room. To maintain hemodynamic stability, 5 mg ephedrine was administered when the mean arterial pressure decreased below 60 mmHg. Red blood cells were transfused if the hemoglobin concentration decreased to <7 g/dL in the perioperative period. 

All hepatic resections were performed by one surgeon using the same hepatic parenchymal transection technique. The extent of hepatic resection was determined based on tumor size and location. Parenchymal transection was performed using an ultrasonic aspirator, metal clips, and electrocautery device, and the cutting surface of the liver was sprayed with biological glue. 

### 2.4. Statistical Analysis 

Clinico-demographic characteristics of patients, operative outcomes of patients and differences in operative outcomes according to BMI and CVP were compared using Student’s *t*-test for distributed data, presented as means ± standard deviations and χ2 test for descriptive data. Univariate and multivariate analysis of risk factors for EBL ≥ 500 mL were performed using an ordinary logistic regression model. *p* values < 0.05 were considered to indicate statistical significance. Statistical analysis was conducted using statistical package for the social sciences (SPSS) version 19.0 (IBM Corp., Armonk, NY, USA). 

## 3. Results

### 3.1. Clinico-Demographic Characteristics of Patients

The clinico-demographic characteristics of the patients are summarized in Table 1. The mean BMI was significantly higher in obese group than in non-obese group (28.0 ± 3.1 vs. 22.1 ± 1.7, kg/m^2^, *p* < 0.001). DM (31.0% vs. 13.2%, *p* = 0.004) and HTN (54.8% vs. 30.8%, *p* = 0.001) were more prevalent in obese group compared to those in non-obese group, with statistical significance. However, there were no significant between-group differences regarding age, sex distribution, diagnosis, and baseline liver function including total bilirubin, INR, and albumin. Major resections were more frequently performed than minor resections in both groups, and there was no significant difference between the groups (*p* = 0.760). 

### 3.2. Intraoperative and Postoperative Patient Outcomes

A CVP at the start of surgery was higher in obese group, than in non-obese group, but the difference was not statistically significant (6.8 ± 3.1 vs. 6.4 ± 3.3, *p* = 0.478). The mean operative duration was significantly longer in obese group, compared with non-obese group (195.7 ± 62.9 vs. 176.0 ± 53.6, mins, *p* = 0.027). There were no significant differences of the amount of intraoperative fluid administration including crystalloid or colloid and urine output, requirements of vasopressor or blood transfusion and ICU admission rate between the groups. EBL (587 ± 694 vs. 430 ± 422, mL, *p* = 0.022) and proportion of EBL ≥ 500 mL (61.9% vs. 40.7%, *p* = 0.005) during hepatic resection were significantly increased in the obese group compared to those in the non-obese group. the incidence of overall postoperative complications (47.6% vs. 25.3%, *p* = 0.002) including wound infection (8.3% vs. 1.1%, *p* = 0.022) and abdominal wall hernia (4.8% vs. 0%, *p* = 0.035), were significantly increased in obese group than in non-obese group. There was no mortality in both groups. Postoperative hospital stay was longer in the obese group compared with the non-obese group, with statistical significance (11.7 ± 4.4 vs. 12.4 ± 7.0, days, *p* = 0.044; Table 2).

### 3.3. Risk Factor Analysis for EBL ≥500 mL during Hepatic Resection

Obesity (odds ratio (OR), 2.204; 95% confidence interval (CI), 1.177–4.129, *p* = 0.014) and CVP ≥5 mmHg at start of surgery (OR, 2.733; 95% CI, 1.445–5.170, *p* = 0.002) were the significant predictors for EBL ≥500 mL during hepatic resection in multivariate analysis (Table 3).

### 3.4. Influence of BMI and CVP on Operative Outcomes 

Three groups were classified by their BMI and CVP: obesity with a CVP <5 mmHg (group A, n = 31); obesity with a CVP ≥5 mmHg (group B, n = 53); non-obese patients (group C, n = 91). CVPs during hepatic resection showed similar trends in all groups that they were decreased from the start of surgery until completion of hepatic parenchymal transection and then re-increased after fluid challenge. Group A had significantly lower CVPs from the start of surgery to postoperative 3 hr than the other groups (Figure 1). 

CVP continuously decreases after the start of surgery until completion of hepatic parenchymal transection, and then starts to increase post-surgery in all groups. The width of decrease was different between the groups, such that a significant difference of CVP at postoperative 4 hr was observed (*p* = 0.032).

Operative duration (186 ± 56 vs. 221 ± 74 vs. 197 ± 67, mins, *p* = 0.043; Figure 2A) and EBL (427 ± 287 vs. 681 ± 543 vs. 430 ± 422, ml, *p* = 0.003; Figure 2B) in group A were significantly decreased compared to those in group B and similar to group C. With respect to postoperative complications, the incidence of AKI was significantly increased in group A compared with the other groups (25.8% vs. 7.5% vs. 11.0%, *p* = 0.041; Figure 2C). However, there was no significant difference in postoperative hospital stay between the groups (11.7 ± 4.2 vs. 11.7 ± 4.6 vs. 10.6 ± 2.6, days, *p* = 0.133; Figure 2D).

Estimated blood loss (*p* = 0.003) and operative duration (*p* = 0.009) were significantly decreased in obese patients with CVP <5, compared to those of the other groups. However, there were no significant differences in postoperative hospital stay and incidence of postoperative complication between the groups.

Creatinine levels were significantly increased in group A compared to other groups from preoperative to postoperative day 7 (Figure 3A). A comparison of the daily fluid balances from operative day to postoperative day 7 to identify the differences in fluid management showed no significant differences between the groups (Figure 3B). Daily urine outputs were lower in group A than in other groups from operative day to postoperative day 5; however, the decrease was not significant (Figure 3C).

Creatinine levels were significantly increased in group A compared to other groups from preoperative to postoperative day 7. However, there are no significant differences in the daily fluid balances and urine output between the two groups. 

## 4. Discussion

We found that intraoperative blood loss was significantly increased in obese patients, compared to non-obese patients. This might be related to increments in the ventilation pressure in patients with elevated BMI, which impair hepatic venous outflow and promote bleeding at the time of hepatic parenchymal transection [10,11]. Maintaining a low CVP during hepatic resection would be more important in these patients, which leads to less blood loss and easier control of the bleeding from the hepatic venous injury by decreasing the influence of ventilation pressure requirements [12]. This study showed that EBL was significantly decreased in obese patients with a low CVP at start of surgery compared to those with a high CVP.

Preoperative fluid restriction is one of the most effective and commonly used methods for lowering CVP [17]. In the present study, although all the patients underwent fluid restriction, CVP at start of surgery varied from 1 to 14 mmHg, such that it might be not enough for some patients. Various methods to decrease CVP in patients with elevated CVP in the operating room were reported previously, but were mostly not effective. Changes in the patient’s body position, such as head-up tilt position, significantly decreased CVP; however, they did not decrease hepatic vein pressure [18]. Diuretics or vasodilators can be administered to decrease CVP, but the effectiveness would be limited because these drugs require a long duration for action and liver has a large amount of blood supply from the portal vein [19]. Inferior vena cava clamping could be a direct method to stop troublesome venous bleeding, but is not widely used because of the technical difficulties and hemodynamic instability of patients [20]. Hypovolemic phlebotomy has been proposed to reduce circulating blood volume, but it also decreases splanchnic blood flow and sometimes requires a blood transfusion [21]. A previous report showed that bioelectrical impedance analysis can be used for preoperative volemia assessment with the advantages of noninvasiveness, rapid processing, easy handling, and relatively inexpensive cost [22]. 

Lowering preload during the hepatic resection could decrease cardiac output, and might lead to the hypoperfusion of abdominal organs, making it a risk factor for postoperative renal dysfunction [23]. In this study, standard definitions of biochemical renal dysfunction were applied, such that postoperative AKI was significantly more common in obese patients with a low CVP than other groups. Although biochemical alterations in renal function are relatively common, clinically relevant renal dysfunction was reported to very rarely develop in previous study [8]. This study also showed similar results, with no significant differences in daily urine outputs between the groups, and none of the patients required renal replacement therapy. However, patients with a high BMI showed a prevalence of DM or cardiovascular diseases, which could be another risk factors for the development of postoperative AKI. Large amounts of fluid were administrated after the completion of hepatic parenchymal dissection to prevention renal dysfunction in this study. 

Obesity is a well-known risk factor for wound-related complication after hepatic resection [10]. A previous study showed that respiratory complications, such as pleural effusion and pulmonary atelectasis, were also more frequent in obese patients [7]. There was no mortality after hepatic resection in this study; however, one study reported that extreme obesity (BMI ≥ 40) was an independent risk factor for increased mortality because when complications develop in those patients, they are likely to be severe and potentially life-threatening [11].

There are some limitations to this study. This study was of a retrospective nature, the accuracy of the data analyzed relies on the completeness of the medical records maintained by our hospital. In addition, the study population were relatively small, and thus, further large-scale prospective studies are warranted to clarify the influence of the obesity and fluid balance. 

## 5. Conclusions

EBL in obese patients with a low CVP was significantly decreased compared those with a high CVP and similar to non-obese patients without a clinically relevant renal dysfunction. Therefore, preoperative fluid restriction to lower CVP should be considered to minimize blood loss during hepatic resection in obese patients.

## Figures and Tables

**Figure 1 jpm-12-01897-f001:**
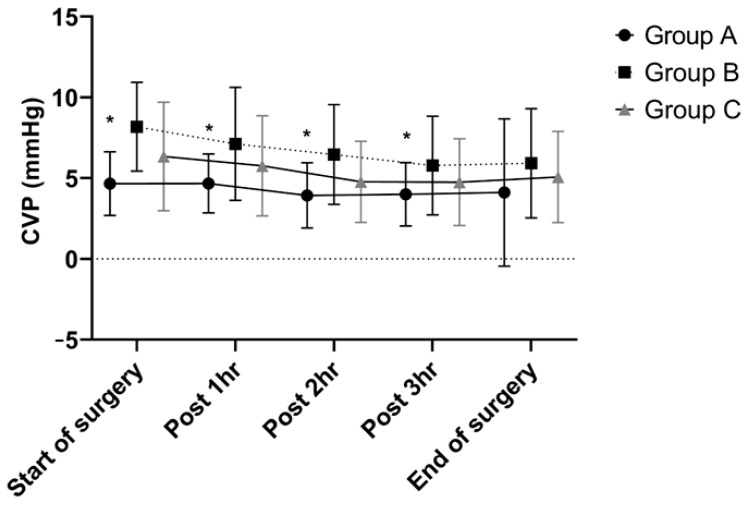
Changes in CVP during hepatic resection. Group A had significantly lower CVPs from the start of surgery to postoperative 3 hr than the other groups. * *p* < 0.05 CVP, central venous pressure.

**Figure 2 jpm-12-01897-f002:**
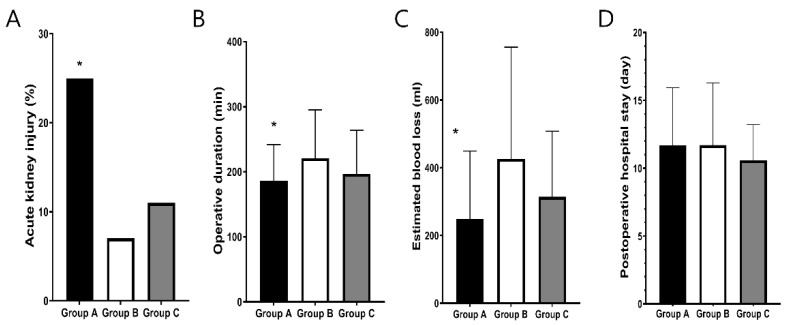
Differences in intraoperative and postoperative outcomes including estimated blood loss (**A**), operative duration (**B**), postoperative hospital stay (**C**), and incidence of postoperative complication (**D**) according to BMI and CVP. * *p* < 0.05 BMI, body mass index; EBL; CVP, central venous pressure.

**Figure 3 jpm-12-01897-f003:**
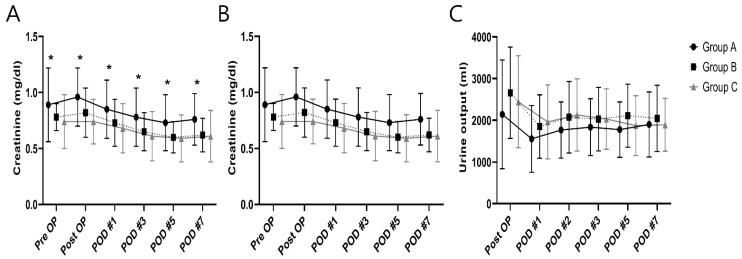
Between-group comparisons in daily serum creatinine (**A**), fluid balances (**B**) and urine outputs (**C**) in the perioperative period. * *p* < 0.05.

**Table 1 jpm-12-01897-t001:** Clinico-demographic characteristics of patients.

	Obese Group (n = 84) (BMI ≥ 25 kg/m^2^)	Non-Obese Group (n = 91) (BMI < 25 kg/m^2^)	*p* Value
Age (year), mean	57.3 ± 15.6	53.7 ± 16.4	0.141
Sex (male)	62 (73.8%)	59 (64.8%)	0.199
BMI	28.0 ± 3.1	22.1 ± 1.7	<0.001
Diabetes mellitus	26 (31.0%)	12 (13.2%)	0.004
Hypertension	46 (54.8%)	28 (30.8%)	0.001
Diagnosis			
Hepatocellular carcinoma	68 (81.0%)	70 (76.9%)	
Cholangiocarcinoma	3 (3.6%)	3 (3.3%)	
Colorectal liver metastasis	8 (9.5%)	12 (13.2%)	
Other benign disease	5 (6.0%)	6 (6.6%)	0.887
Baseline liver function			
TB	0.7 ± 0.3	0.6 ± 0.2	0.679
Albumin	4.2 ± 0.4	4.3 ± 0.4	0.835
INR	1.06 ± 0.07	1.08 ± 0.10	0.122
Procedure, n (%)			
Major resection (≥3 segments)	47 (56.0%)	53 (58.2%)	
Minor resection (<3 segments)	37 (44.0%)	38 (41.8%)	0.760

Abbreviations: BMI, body mass index; TB, total bilirubin; INR, International normalized ratio.

**Table 2 jpm-12-01897-t002:** Operative outcomes of patients.

	Obese (n = 84) (BMI ≥ 25 kg/m^2^)	Non-Obese (n = 91) (BMI < 25 kg/m^2^)	*p* Value
CVP at start of surgery	6.8 (±3.1)	6.4 (±3.3)	0.478
Operative duration, min	195.7 (±62.9)	176.0 (±53.6)	0.027
Intraoperative fluids			
Crystalloid, mL	2127 (±1016)	2087 (±977)	0.790
Colloid, mL	282 (±292)	253 (±271)	0.497
Urine output, mL	329 (±276)	327 (±291)	0.960
Vasopressor usage	19 (20.9%)	9 (15.5%)	0.414
Blood transfusion	11 (13.1%)	9 (9.9%)	0.506
EBL, mL	587 (±694)	430 (±422)	0.022
EBL ≥ 500 mL	52 (61.9%)	37 (40.7%)	0.005
Postoperative complications	40 (47.6%)	23 (25.3%)	0.002
Wound infection	7 (8.3%)	1 (1.1%)	0.022
Pneumonia	2 (2.4%)	0	0.139
AKI	12 (14.3%)	10 (11.0%)	0.511
Liver insufficiency	14 (16.7%)	11 (12.1%)	0.387
Abdominal wall hernia	4 (4.8%)	0	0.035
Acute myocardial infarction	1 (1.2%)	0	0.297
ICU admission	12 (14.3%)	11 (12.1%)	0.667
Mortality	0	0	−
Postoperative hospital stay, days	11.7 (±4.4)	10.6 (±2.6)	0.044

Data are presented as the mean ± standard deviations or n (%) unless otherwise indicated. CVP, central venous pressure; EBL, estimated blood loss; AKI, acute kidney injury; ICU, intensive care unit; BMI, body mass index.

**Table 3 jpm-12-01897-t003:** Analysis of risk factors for estimated blood loss ≥ 500 mL.

	Univariate Analysis			Multivariate Analysis		
Variable	OR	95% CI	*p* Value	OR	95% CI	*p* Value
Age (year)	1.001	0.983–1.020	0.892			
Sex (male)	1.451	0.761–2.765	0.258			
Obesity (BMI ≥ 25 kg/m^2^)	2.372	1.298–4.353	0.005	2.204	1.177–4.129	0.014
Diabetes mellitus	0.957	0.466–1.964	0.905			
Hypertension	0.781	0.428–1.425	0.420			
TB	2.774	0.827–9.301	0.098			
INR	0.866	0.032–23.674	0.932			
Albumin	0.844	0.395–1.818	0.665			
Major resection	1.342	0.736–2.445	0.337			
CVP ≥ 5 mmHg at start of surgery	2.787	1.494–5.196	0.001	2.733	1.445–5.170	0.002

Abbreviations: BMI, body mass index; TB, total bilirubin; INR, International normalized ratio; CVP, central venous pressure; OR, odds ratio; CI, confidence interval.

## Data Availability

Data available from the author upon request.

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
