# Peer review of "Influence of Obesity and Fluid Balance on Operative Outcomes in Hepatic Resection"

_jpm, 2022, doi:10.3390/jpm12111897_

Round 1

Reviewer 1 Report

Manuscript ID: jpm-2016831

This article could be provided information for decision making in the restriction fluid balance before hepatic resection in obese patients and this way improving it operative outcomes, however, I have some questions about this manuscript

Comments

Line 24 you start talking about hepatocellular carcinoma, however, in your title, keywords and in the objective of the study it is not mentioned, I considered that is necessary to start your introduction with hepatic resection and then on the implication of hepatocellular carcinoma in obese patients.

Line 32 add in this section, what is the prevalence in obese people for developing colorectal liver metastasis? why the high body mass index stimulates the appearance colorectal liver metastasis?

Line 34-35 you mentioned the efforts at understanding risk of operative morbidity and mortality, I think is important understand if the fluid balance could be restricted or not, before a surgery intervention and this information could be to help predictive of morbidity and mortality rates.

Line 65-66 can you add patients’ number for each group of central venous pressure (CVP) at start of surgery?

Line 111-112 you mention in statistical analysis that you used Student´s t or Kruskal-Wallis test between group comparisons, however, could be used Student´s t for parametric analysis or Mann-Whitney for non-parametric analysis. if you used Kuskal-Wallis, maybe you should have used One-Way ANOVA for parametric analysis, could you mention more about this? Could you add the multivariate analysis in this section?

Line 128-129 could you highlight the significance values in table 1? (BMI, diabetes mellitus and hypertension).

Line   In the section 2.1 (patients) you mention only 3 groups non obese group, obesity with a CVP <5 mmHg and obesity with a CVP ≥5 mmHg, however, in the section 3.2 you mentioned one group more whit CVP at the start of surgery in non-obese group. could you explain or correct this information please?

Line 145-146 could you highlight the significance values in table 2 and 3?

Line 158-160 in section 3.2 your mention CVP non-obese group and in this section, you write only 3 groups (non-obese group, obesity with a CVP <5 mmHg; obesity with a CVP ≥5 mmHg, maybe you can explain lack of a group.

Line 165-166 can you add a little description of the figure 1?

Line 201-211 these paragraphs should be in introduction section. In the discussion section, you could start talking about hepatic resection.

Line 214 what are the other patients’ groups?

Line 251-256 in these paragraphs you can remove “This study showed that wound infection and abdominal wall hernia were significantly increased in obese patients. It might be also related to a higher prevalence of DM in obese patients that glycemic control during the perioperative period would be important for reducing these complications. Preoperative respiratory rehabilitation exercise, early postoperative ambulation, and pain control for expectoration of sputa might would be helpful for its prevention”.

Line 275-274 this is the first conclusion according to the manuscript title and the main objective.

Author Response

This article could be provided information for decision making in the restriction fluid balance before hepatic resection in obese patients and this way improving it operative outcomes, however, I have some questions about this manuscript

Ans) I’m thank the reviewer for going through our manuscript carefully and suggesting points to improve the same.

Comments

Line 24 you start talking about hepatocellular carcinoma, however, in your title, keywords and in the objective of the study it is not mentioned, I considered that is necessary to start your introduction with hepatic resection and then on the implication of hepatocellular carcinoma in obese patients.

Ans) I revised the start of introduction considering your opinion. I changed the order of sentences as below.

‘Recent advances in perioperative care, surgical techniques, and dissection devices expand the indications for hepatic resection that operation for patients with advanced HCC and metastatic tumor on the liver are on the rise [1,2]. Wide spread of hepatocellular carcinoma (HCC) surveillance in high-risk populations, such as cirrhotic patients, also has led to an increase in detection of early-stage HCC that is eligible for hepatic resection [3].’ (L26-30)

Line 32 add in this section, what is the prevalence in obese people for developing colorectal liver metastasis? why the high body mass index stimulates the appearance colorectal liver metastasis?

Ans) This comment might be confusing. The meaning of this comment was that obesity is a significant risk factor for colorectal cancer that it also has an influence on the increment of colorectal liver metastasis. I removed this comment.

Line 34-35 you mentioned the efforts at understanding risk of operative morbidity and mortality, I think is important understand if the fluid balance could be restricted or not, before a surgery intervention and this information could be to help predictive of morbidity and mortality rates.

Ans) I agree with your opinion that I erased this comment in the manuscript. And the above comments were included in the aim of the study in the introduction section.

Line 65-66 can you add patients’ number for each group of central venous pressure (CVP) at start of surgery?

Ans) I added patients’ number for each group by BMI and CVP at start of surgery.

‘obesity with a CVP <5 mmHg (group A, n =31); obesity with a CVP ≥5 mmHg (group B, n=53); non-obese patients (group C, n=91).’ (L70-71)

Line 111-112 you mention in statistical analysis that you used Student´s t or Kruskal-Wallis test between group comparisons, however, could be used Student´s t for parametric analysis or Mann-Whitney for non-parametric analysis. if you used Kuskal-Wallis, maybe you should have used One-Way ANOVA for parametric analysis, could you mention more about this? Could you add the multivariate analysis in this section?

Ans) There were some mistakes that Kruskal-Wallis test was not used in this study. And I used an ordinary logistic regression model for univariate and multivariate analysis of risk factors for EBL ≥ 500 mL in this study. I revised this section that more detailed method of statistical analysis was mentioned as below.

‘Clinico-demographic characteristics of patients, operative outcomes of patients and differences in operative outcomes according to BMI and CVP were compared using Stu-dent’s t-test for distributed data, presented as means ± standard deviations and χ2 test for descriptive data. Univariate and multivariate analysis of risk factors for EBL ≥ 500 mL were performed using an ordinary logistic regression model. p values < 0.05 were consid-ered to indicate statistical significance. Statistical analysis was conducted using statistical package for the social sciences (SPSS) version 19.0 (IBM Corp., Armonk, NY, USA).’ (L112-119)

Line 128-129 could you highlight the significance values in table 1? (BMI, diabetes mellitus and hypertension).

Ans) I highlight the significant variables using bold type in Table 1.

Line   In the section 2.1 (patients) you mention only 3 groups non obese group, obesity with a CVP <5 mmHg and obesity with a CVP ≥5 mmHg, however, in the section 3.2 you mentioned one group more whit CVP at the start of surgery in non-obese group. could you explain or correct this information please?

Ans) At first, in the section 3.2, we compared the clinical outcomes of two groups (obese and non-obese group). And then, we compared three groups (non-obese group, obesity with a CVP <5 mmHg; obesity with a CVP ≥5 mmHg), analysed in the section 3.4.

Line 145-146 could you highlight the significance values in table 2 and 3?

Ans) I highlight the significant variables using bold type in Table 2 and Table 3.

Line 158-160 in section 3.2 your mention CVP non-obese group and in this section, you write only 3 groups (non-obese group, obesity with a CVP <5 mmHg; obesity with a CVP ≥5 mmHg, maybe you can explain lack of a group.

Ans) At first, in the section 3.2, we compared the clinical outcomes of two groups (obese and non-obese group). And then, we compared three groups (non-obese group, obesity with a CVP <5 mmHg; obesity with a CVP ≥5 mmHg), analysed in the section 3.4.

Line 165-166 can you add a little description of the figure 1?

Ans) I added a little description of the figure 1 as below.

‘Group A had significantly lower CVPs from the start of surgery to postoperative 3 hr than the other groups’

Line 201-211 these paragraphs should be in introduction section. In the discussion section, you could start talking about hepatic resection.

Ans) I also agree with your opinion. I removed these paragraphs.

Line 214 what are the other patients’ groups?

Ans) This comment might be confusion that I changed the other patients to non-obese patients.

Line 251-256 in these paragraphs you can remove “This study showed that wound infection and abdominal wall hernia were significantly increased in obese patients. It might be also related to a higher prevalence of DM in obese patients that glycemic control during the perioperative period would be important for reducing these complications. Preoperative respiratory rehabilitation exercise, early postoperative ambulation, and pain control for expectoration of sputa might would be helpful for its prevention”.

Ans) I removed these paragraphs considering your opinion.

Line 275-274 this is the first conclusion according to the manuscript title and the main objective.

Ans) I revised the conclusion section considering your opinion as below.

‘EBL in obese patients with a low CVP was significantly decreased than those with a high CVP and similar with non-obese patients without a clinically relevant renal dysfunc-tion. Therefore, preoperative fluid restriction for lowering a CVP would be considered for minimizing blood loss during hepatic resection in obese patients.’ (L255-258)

Reviewer 2 Report

Manuscript jpm-2016831 suggestions are listed below.

Figure 1 define the groups in the figure, only group C with the asterisk is identified

In table 3 results specify that the statistically significant results have large confidence intervals due to the stratification of the sample or sample size is small.

Line 235 and 236 to improve the wording of this paragraph, it is confused since it begins ¨ We previously...¨ it seems that they are the results of this study

Author contribution, says All authors... refer to singular since it is the only author

References 20 and 14 unify the style, are found with uppercase and lowercase letters

Author Response

Manuscript jpm-2016831 suggestions are listed below.

Ans) I’m thank the reviewer for going through our manuscript carefully and suggesting points to improve the same.

Figure 1 define the groups in the figure, only group C with the asterisk is identified

Ans) Group A had significantly lower CVPs from the start of surgery to postoperative 3 hr than the other groups. I added the asterisk in group A to highlight the significance.

In table 3 results specify that the statistically significant results have large confidence intervals due to the stratification of the sample or sample size is small.

Ans) This study was a retrospective nature and the study population were relatively small that further large-scale prospective studies are warranted in order to clarify the influence of the obesity and fluid balance. However, this study can provide information for decision making in the restriction fluid balance before hepatic resection in obese patients and this way improving it operative outcomes.

Line 235 and 236 to improve the wording of this paragraph, it is confused since it begins ¨ We previously...¨ it seems that they are the results of this study

Ans) I agree with your opinion that this comment is confusion. I revised the comment as below.

‘Previous report showed that bioelectrical impedance analysis can be used for preoperative volemia assessment with the advantages of noninvasiveness, rapid processing, easy handling, and relatively inexpensive cost’ (L227-229)

Author contribution, says All authors... refer to singular since it is the only author

Ans) It seems that the comment of author contribution is not necessary. I removed the comments.

References 20 and 14 unify the style, are found with uppercase and lowercase letters

Ans) I correct the different style of references in the manuscript.

Reviewer 3 Report

Thank you. The authors said obese patients with a low CVP was significantly decreased than those with a high CVP and similar with non-obese patients without a clinically relevant renal dysfunction.  I think this paper was few data used few peoples. you should analyze more experiments in obese patients and non-obese patients.  Please continue to do more analysis and show more interested data.  

Author Response

Thank you. The authors said obese patients with a low CVP was significantly decreased than those with a high CVP and similar with non-obese patients without a clinically relevant renal dysfunction.  I think this paper was few data used few peoples. you should analyze more experiments in obese patients and non-obese patients. Please continue to do more analysis and show more interested data.  

Ans) We thank the reviewer for going through our manuscript carefully. We agree with your opinion. This study was a retrospective nature and the study population were relatively small that further large-scale prospective studies are warranted in order to clarify the influence of the obesity and fluid balance. However, this study can provide information for decision making in the restriction fluid balance before hepatic resection in obese patients and this way improving it operative outcomes. We added comments about the limitation of this study in the discussion section.

‘There are some limitations to this study. This study was a retrospective nature, the accuracy of the data analyzed relies on the completeness of medical records maintained by our hospital. In addition, the study population were relatively small, and thus, further large-scale prospective studies are warranted in order to clarify the influence of the obesity and fluid balance.’

Round 2

Reviewer 1 Report

The text has to minor editing errors

Reviewer 3 Report

Thank you to select me as a reviewer. I have no claim and accepted this paper.